# Food-Protein-Induced Proctocolitis in Pre-Term Newborns with Bloody Stools in a Neonatal Intensive Care Unit

**DOI:** 10.3390/nu16173036

**Published:** 2024-09-09

**Authors:** Enza D’Auria, Francesco Cavigioli, Miriam Acunzo, Paola Azzurra La Verde, Anna Di Gallo, Carolina Piran, Lodovico Sterzi, Gian Vincenzo Zuccotti, Gianluca Lista

**Affiliations:** 1Department of Pediatrics, Buzzi Children’s Hospital, 20154 Milan, Italy; enza.dauria@asst-fbf-sacco.it (E.D.); anna.digallo@unimi.it (A.D.G.); carolina.piran@studenti.unimi.it (C.P.); gianvincenzo.zuccotti@unimi.it (G.V.Z.); 2Division of Neonatology and Neonatal Intensive Care Unit, Buzzi Children’s Hospital, 20154 Milan, Italy; francesco.cavigioli@asst-fbf-sacco.it (F.C.); paola.laverde@asst-fbf-sacco.it (P.A.L.V.); gianluca.lista@asst-fbf-sacco.it (G.L.); 3Department of Biomedical and Clinical Sciences, Pediatric Clinical Research Center “Romeo and Enrica Invernizzi”, University of Milan, 20157 Milan, Italy; lodovico.sterzi@unimi.it; 4Department of Biomedical and Clinical Sciences L. Sacco, University of Milan, 20154 Milan, Italy

**Keywords:** hematochezia, pre-term infants, NEC, food allergy, proctocolitis, CMPA

## Abstract

The bloody stools of newborns may be a clue for several clinical entities of varying severity, ranging from idiopathic neonatal transient colitis to food-protein-induced allergic proctocolitis (FPIAP) or necrotizing enterocolitis (NEC). Distinguishing among them at an early stage is challenging but crucial, as the treatments and prognoses are different. We conducted a monocentric retrospective study including all pre-term infants with bloody stools admitted to the Neonatal Intensive Care Unit (NICU) of the Vittore Buzzi Children’s Hospital (Milan) from December 2022 to May 2024. Patients diagnosed with NEC exhibited significantly lower eosinophil counts and higher procalcitonin levels than both patients with FPIAP and patients with idiopathic neonatal transient colitis, as well as a statistically significant increase in pathological features from abdomen ultrasounds and abdominal X-rays. In contrast, no lab markers or imaging techniques have been demonstrated to be useful in distinguishing between idiopathic neonatal transient colitis and FPIAP. Thus, after excluding a diagnosis of NEC, the only way to confirm FPIAP is through the oral food challenge, which can be performed in premature newborns presenting with bloody stools who are otherwise healthy and under medical supervision, in order to identify infants who may benefit from a cow’s-milk-free diet.

## 1. Introduction

The bloody stools of newborns may be a sign of several clinical entities of different degrees of severity. It may range from an idiopathic neonatal transient colitis to food-protein-induced allergic proctocolitis (FPIAP) to more severe diseases, such as necrotizing enterocolitis (NEC).

Distinguishing among them at an early stage is challenging but crucial, as the treatments and prognoses are radically different. NEC is a life-threatening condition that must be treated with a period of fasting and antibiotic therapy and, in the most severe cases, with intestinal resection surgery. On the other hand, food-protein-induced allergic proctocolitis is usually a benign condition that does not require pharmacological treatment, but only the avoidance of cow’s milk protein.

FPIAP is part of a larger group of gastrointestinal disorders associated with non-IgE-mediated food allergies (non-IgE-GI-FA). It is caused by inflammation of the distal portion of the sigma and rectum characterized by edema and erosion of the mucosa with eosinophilic infiltration of the epithelium and/or lamina propria. Onset most frequently occurs in the first three months of life, and it is manifested by blood in the feces (hematochezia or occult blood) and is less frequently associated with half-formed stools and mucus. Typically, infants with FPIAP appear healthy, with normal weight growth, and have no other associated symptoms, except for mild anemia in the case of chronic bleeding [1].

The prevalence is not well known and has been estimated at between 0.16% and 64% of healthy infants with hematochezia. Cow’s milk protein is the allergen most frequently responsible for FPIAP.

More than 50% of infants with FPIAP are exclusively breast-fed, and up to 25% of cases have a positive family history of atopy. In breast-fed infants, symptoms often occur later than in formula-fed infants [2]. It is possible that symptoms begin as early as the first week after birth, suggesting that in utero sensitization may occur, although the exact mechanism is not yet fully understood [3].

The main treatment of FPIAP is to avoid the allergen responsible—in most cases, cow’s milk protein. In breast-fed, symptomatic infants, a cow’s milk protein exclusion diet may be recommended for mothers, especially in cases of moderate/severe bleeding. In infants who continue to be symptomatic despite the mother’s strict cow’s milk protein exclusion diet, the elimination of other allergens, such as soy and eggs, from the mother’s diet should be considered (rarely other foods) [2].

Visible bright red rectal bleeding, also associated with eosinophilic infiltration in the rectosigmoid mucosa, may disappear spontaneously in infants who have been excluded from surgical, infection, and other pathologies [4,5].

Hence, a “watch and wait” approach of 1 month before an elimination diet is recommended in order to monitor whether spontaneous resolution occurs in some patients [6].

The onset of FPIAP symptoms usually occurs in the first months of life; the majority of patients develop symptoms during the second month of life [7,8].

However, the large cohort study by Senocak et al. found that symptoms began in the neonatal period in one third of patients, finding an association between the onset of symptoms in the neonatal period and a history of premature birth [8], although the available data in pre-term newborns is scarce and heterogeneous.

Cases of cow’s milk protein allergy (CMPA) reported in pre-term infants are mostly non-IgE mediated [3]; Morita et al. found an incidence of CMPA with systemic symptoms of 1.1%, although to date, the prevalence is not well established [9].

Most pre-term infants with CMPA were exposed to bovine milk protein before the development of symptoms, and the most common clinical manifestations were bloody stools, vomiting, and abdominal distension [10]. CMPA in pre-term infants may present with a more acute illness that could appear to mimic NEC, sepsis, or shock [11,12], or they may have non-specific symptoms, including abdominal discomfort and fever [13].

The aim of the present study was to describe the features of pre-term newborns with bloody stools in a neonatal intensive care unit, specifically focusing on newborns with a final diagnosis of FPIAP.

## 2. Materials and Methods

We conducted a monocentric retrospective study in a neonatal intensive care unit (NICU). The study population consisted of all pre-term newborns with bloody stools admitted to the NICU of the Vittore Buzzi Children’s Hospital in Milan from December 2022 to May 2024.

At the onset of the symptoms, all subjects were fasted for 48 h, and they underwent a diagnostic evaluation including blood examination, abdominal X-rays (single or serial, as appropriate), abdominal ultrasound, and fecal infection tests.

The diagnosis of NEC was based on Bell’s modified criteria (Walsh and Kliegman, 1986) [14], which include clinical and radiological criteria.

Patients for whom the diagnosis of NEC was ruled out were re-fed with amino acid formula and after one week, a challenge was carried out to confirm or rule out a diagnosis of CMPA.

Newborns whose symptoms relapsed (e.g., presence of bloody stools after cow’s milk reintroduction) were diagnosed as having a cow’s milk allergy, and they continued to be fed with the aminoacidic formula, while infants who had no recurrence of symptoms upon the reintroduction of CMP in the diet were diagnosed with idiopathic neonatal transient colitis.

The following clinical and laboratory parameters were collected from the medical records: date of birth; sex; gestational age (GA); twin pregnancy (specifying whether monochorionic or bichorionic); type of delivery; birth weight; age at onset of hematochezia; presence of other symptoms (abdominal distension, vomiting, and any other); type of feeding undertaken in the first few days of life; type of feeding in the days preceding the episode; blood exams (blood count, IgE, sIgE, blood culture, fecal antigens of adenovirus and rotavirus, fecal norovirus and enterovirus, X-rays of the abdomen, ultrasound of the abdomen, and allergy counseling); duration and type of antibiotic therapy carried out; duration of withdrawal of enteral feeding; and type of feeding undertaken when the episode was resolved.

Data analysis was conducted using R software version 4.1.2. Quantitative and categorical variables available as metadata were compared among patients diagnosed with CMPA, idiopathic neonatal transient colitis, and NEC. Quantitative variables were compared between groups using the Kruskal–Wallis test. Once the overall differences among the groups had been identified using the Kruskal–Wallis test, post hoc pairwise comparisons were performed to pinpoint specific differences between each pair of groups. The Mann–Whitney U test, another non-parametric test, was used for these pairwise comparisons. The results were visualized with box plots, showcasing the median, quartiles, and potential outliers for each group.

Categorical variables were compared between groups using the Chi-squared test, with the results visualized using bar plots. Statistical significance was defined as *p*-values < 0.05.

## 3. Results

A total of 43 prematures showed hematochezia during the study period.

In total, 40% (18/43) of infants with hematochezia were finally diagnosed as NEC, 37.2% (16/43) with FPIAP, and 20% (9/43) with idiopathic neonatal transient colitis.

In total, 11.6% (5/43) were extremely pre-term newborns (<28 weeks’ GA), and none among them were diagnosed with CMPA, while 2/5 presented with idiopathic neonatal transient colitis. As many of 32% (14/43) were very pre-term (28 + 0–31 + 6 weeks’ GA), 50% (7/14) had a diagnosis of FPIAP, and 2/14 (14.2%) presented with idiopathic neonatal transient colitis. As many as 53% (23/43) were moderate-to-late pre-term infants (32 + 0–36 + 6 weeks’ GA), and 40% (9/23) received a diagnosis of FPIAP, while 21% (5/23) had idiopathic neonatal transient colitis.

Two moderate pre-term infants with an initial diagnosis of NEC were later diagnosed with CMPA.

The descriptive characteristics and laboratory parameters of the three study groups are summarized below (Table 1 and Table 2).

The majority of pre-term newborns with FPIAP were exposed to bovine-based milk proteins before the development of symptoms. At the time of the onset of symptoms, 5/16 (31.3%) with FPIAP infants were fed with breast milk, 2/16 (12.5%) with formula milk, and 9/16 (56.3%) with mixed feeding. A total of 5/16 infants were receiving fortified breast milk.

In 93.7% (15/16) of the cases, patients with FPIAP presented sIgE-negative and were diagnosed with allergic proctocolitis. Only one patient (35.4 weeks’ GA) had total IgE 14.5 KU/L, alpha lactalbumin < 0.1 kU/L, beta lactoglobulin 2.31 kU/L, and casein 0.41 kU/L and was the only one to present with vomiting, abdominal distension, and gastric stagnation. Of the remaining 15 patients, only 1 (28 + 3 weeks’ GA) presented with symptoms of gastroesophageal reflux and dermatitis in addition to rectal bleeding.

The statistical analyses of the clinical data revealed some features associated with CMPA, idiopathic neonatal transient colitis, and NEC diagnoses. The Kruskal–Wallis test showed that the eosinophil count (*p* = 0.025) and the procalcitonin levels (*p* = 0.033) were significantly different among the groups. As shown in Figure 1, post hoc pairwise comparison via Mann–Whitney U test revealed that patients diagnosed with NEC exhibited a significantly lower eosinophil count compared to both patients with CMPA (*p* = 0.016) and patients with transient hematochezia (Kruskal–Wallis test, *p* = 0.035).

Although not reaching full statistical significance in the Kruskal-Wallis test, blood platelet counts and C-reactive protein levels showed trends of decrease and increase, respectively, in patients diagnosed with NEC (Figure 2). The receiver operating characteristic (ROC) curve of eosinophils count between patients with NEC and CMPA/transient hematochezia had an AUC of 81.2% (CI 63.5–99.0%). The optimal threshold was defined as 330, with a sensibility of 81.8%, a specificity of 75%, a positive predictive value of 69%, and a negative predictive value of 86% (Figure 3).

Despite the Kruskal–Wallis *p*-value (*p* = 0.097) not fully supporting the claim, the Mann–Whitney test *p*-value (*p* = 0.05) and visual observation of Figure 2 revealed an interesting trend of increasing neutrophil levels in patients with CMPA compared to those with idiopathic neonatal transient colitis.

Moreover, regarding imaging, pathological outcomes at ultrasounds (*p* = 0.046) and abdominal X-rays (*p* = 0.000096) were statically more frequent in newborns with a final diagnosis of NEC.

## 4. Discussion

The main finding of our study is that neither clinical characteristics nor laboratory markers are helpful in distinguishing between idiopathic neonatal transient colitis and FPIAP in premature newborns with bloody stools.

Nevertheless, lab parameters and imaging, if combined to the clinical features, allow for the exclusion of NEC in prematures, as we previously demonstrated for premature newborns with FPIES [12]. In this regard, the diagnosis of NEC, which needs to be ruled out first in prematures, is mainly a clinical diagnosis, relying on clinical features and a clinical course.

It is noteworthy that the eosinophil count was significantly lower in NEC compared to both idiopathic neonatal transient colitis and FPIAP. It should be considered that eosinophilia is common in prematures, reflecting hematopoiesis [15], and that its incidence increases with low GA and small birth weight [16,17]; however, we did not observe a statistical difference for gestational age and number of days since birth at symptom onset among groups. Furthermore, the results showed that both gestational age (Pearson r = 0.13, *p* = 0.49) and the number of days since birth (Pearson r = −0.15, *p* = 0.43) do not have a significant inverse correlation, suggesting that the eosinophil levels in our study are not influenced by gestational age.

Regarding imaging, abnormal findings—e.g., intestinal pneumatosis—were significantly higher in newborns with a confirmed diagnosis of NEC, in agreement with the literature data.

Intestinal pneumatosis is considered to be a pathognomonic finding of NEC [18]; however, in recent years, it has been shown that such a radiological finding has been observed to occur in different clinical entities, such as in food-protein-induced enterocolitis (FPIES) [12,19].

Intestinal pneumatosis has also been described in FPIAP, in agreement with our findings [20].

Of note, in our study, two newborns who showed intestinal pneumatosis were first diagnosed as NEC, and they were treated accordingly. However, their symptoms relapsed when enteral feeding with a cow’s milk-based formula was reintroduced and then disappeared when they started a CMP-free diet, suggesting a final diagnosis of FPIAP.

Our findings are similar to those observed in a retrospective study on 348 infants in a NICU who required parenteral nutrition. A total of 18/348 (5%) received a final diagnosis of CMPA, manifested by feeding intolerance or “late-onset or recurrent NEC-like illness” [21].

However, we should not overlook the 37% of patients diagnosed with CMPA who presented with pathologic radiography.

As far as gestational age, Morita et al. showed no diagnosis of CMPA < 32 weeks’ GA; conversely, in our study, seven patients < 32 weeks’ GA were diagnosed with CMPA, while no patient < 28 weeks’ GA received a diagnosis of CMPA. There is still a limited understanding of the development of the immune system in pre-term infants; in some studies, an association between prematurity and reduced functionality of Tregs has been shown, compared to full-term infants of adequate weight [22]. Human Tregs are a heterogeneous cellular subgroup in which functionally and phenotypically different subpopulations can be distinguished. Recent studies in a neonatal animal model have reported a delay in migration and ontogeny of Tregs in the intestinal tract and a reduced proportion of Foxp3^+^ Tregs in the intestinal mucosa, which may correlate with gestational age [23,24]. Reduction in regulatory T cells in pre-term newborns is associated with necrotizing enterocolitis [25]. Moreover since SGA neonates show reduced suppressive activity of Tregs [26], this could be an additional factor in promoting the development of FPIAP.

The pathogenesis of FPIAP is not well understood. Increased levels of pro-inflammatory cytokines, an altered permeability of the gut epithelium, which may lead to inflammation and an immune reaction to otherwise benign food proteins, have been hypothesized [27]. However, the exact pathogenetic mechanisms of this disease remain unknown.

In our cohort, all patients underwent an oral food challenge (OFC), which allowed us to diagnose a case of idiopathic neonatal transient colitis in the group of prematures whose symptoms did not relapse after the reintroduction of cow’s milk proteins into the diet.

Making a correct diagnosis and distinguishing between these two entities is important to avoid prolonged and unnecessary diets in patients with idiopathic neonatal transient colitis.

To this end, it should be highlighted that idiopathic neonatal transient colitis is often misdiagnosed as FPIAP, despite biopsy, without OFC testing in full-term infants [5,28]. On the other hand, a missed diagnosis of CMPA in early life can lead to potential complications, such as mild anemia, associated with persistent bleeding, and prolonged colonic inflammation, which, in turn, may predispose the patient to the development of functional gastrointestinal disorders in childhood [29]. All pre-term newborns with hematochezia must be considered at risk of serious diseases (NEC or other surgical conditions such as malrotation, volvulus, Hirschsprung’s disease with enterocolitis, upper gastrointestinal hemorrhaging, vascular malformations, gastrointestinal duplication), and investigations must be performed to exclude these diagnoses first. Once NEC and surgical conditions have been excluded, rectal bleeding is often labeled as CMPA in routine clinical practice. However, it must be borne in mind that within this group hides another, that of idiopathic neonatal transient colitis, which, if undiagnosed, runs the risk of patients receiving an unnecessary and prolonged diet. Clinical presentation, lab markers and/or imaging techniques are not useful in distinguishing between these two conditions; thus, an early oral provocation test remains the only way to exclude the diagnosis of FPIAP.

Recently, a potential marker (IL-27) has been proposed to distinguish between early-onset FPIES and NEC [30], and biomarkers for inflammatory bowel diseases are also currently being investigated in ongoing studies [31,32,33].

Future investigations should be advanced to identify useful diagnostic biomarkers for FPIAP. Currently, an oral provocation test is still necessary to confirm or rule out the diagnosis of FPIAP. Based on our findings, the OFC may be safely performed, under medical supervision, even in premature newborns presenting with bloody stools who are otherwise healthy.

Regarding the mode of feeding, it should be pointed out that premature and very-low-birth-weight infants have additional caloric requirements to human milk that are met through the use of human milk fortificants, which are traditionally CMP-based. Recently, fortificants based on human milk protein have been proposed in the U.S. [34]. In our case series, five patients with CMPA had CMP-based fortificants added to their diet. It is still unknown if the use of these supplements in the first days of life plays a role in the development of CMPA, and this question should be to be further investigated in the future.

This study has some limitations pertaining to the retrospective design and the relatively small number of patients. The strength of this study lies in having performed oral provocation tests in pre-term newborns after a short period of elemental formula.

Further prospective studies are warranted to better elucidate FPIAP in pre-term newborns and its management.

## 5. Conclusions

Our study highlights the importance of considering CMPA when taking care of otherwise-healthy pre-term newborns with bloody stools. Both serial pathological X-ray and ultrasound findings of the abdomen, associated with high serum PCT levels and low eosinophil count, help physicians to make a diagnosis of NEC. In this particularly fragile population, an early oral food challenge could be considered in order to differentiate between idiopathic neonatal transient colitis and FPIAP. Further studies on larger populations are needed to confirm these findings.

## Figures and Tables

**Figure 1 nutrients-16-03036-f001:**
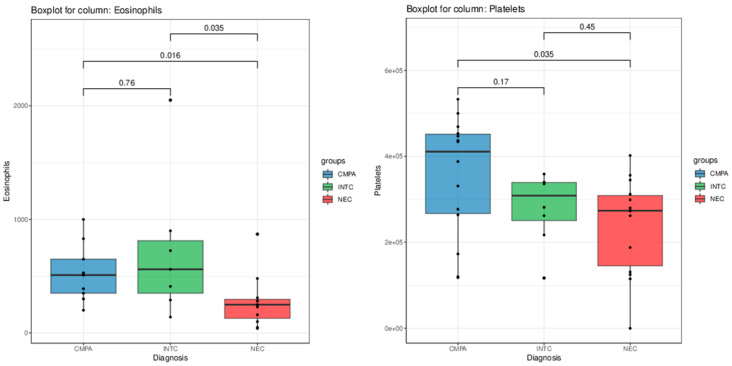
Boxplots of the eosinophil (**on the left**) and platelet (**on the right**) levels in the patient population. The *p*-values of pairwise comparisons between the groups performed via Mann–Whitney U test are shown on top.

**Figure 2 nutrients-16-03036-f002:**
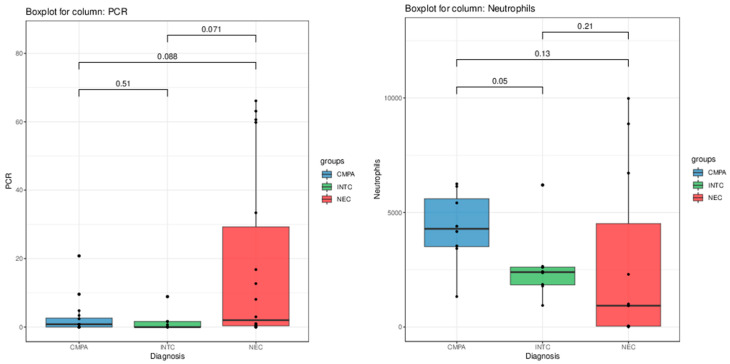
Boxplots of PCR (**on the left**) and neutrophils levels (**on the right**) in the patient population. The *p*-values of pairwise comparisons between the groups performed via Mann–Whitney U test are shown on top.

**Figure 3 nutrients-16-03036-f003:**
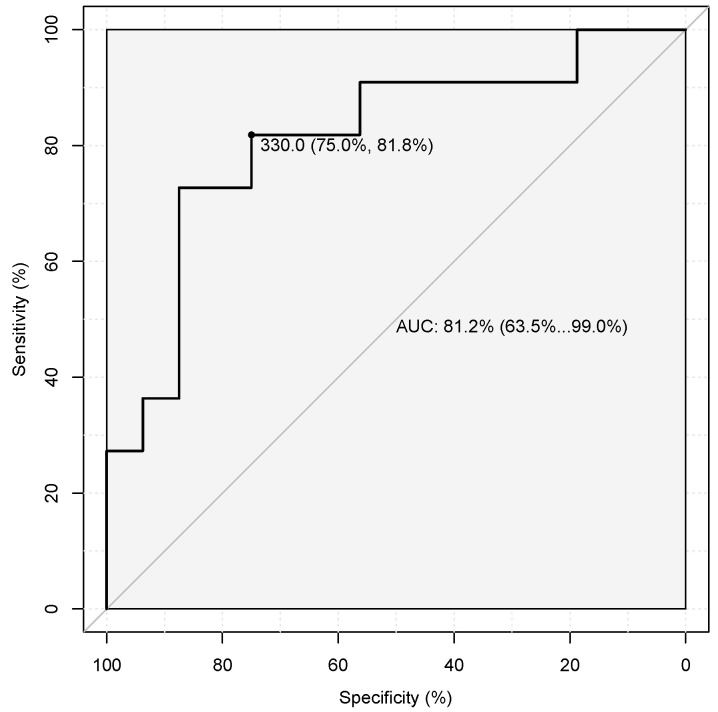
ROC curve of eosinophils.

**Table 1 nutrients-16-03036-t001:** Characteristics of newborns with NEC, FPIAP, and idiopathic neonatal transient colitis.

	FPIAP (16)	Idiopathic Neonatal Transient Colitis (9)	NEC (18)
Mean gestational age (weeks)	34.38 ± 3.19	35.1 ± 4.47	31 ± 4
Twin pregnancies	8 (50%) (4/8 BC, 4/8 MC)	6	12 (66%)
Birth weight	1680 g	1539 g	1606
Cesarean Section	13 (81.3%)	7 (77%)	17 (94%)
Days of onset	16	27	11.1
SGA/IUGR	6 (37.5%)	2 (22%)	3 (16%)
FEF before hematochezia	7 (43.8%)	4 (44%)	6 (33%)
Family history for allergies	5 (31.2%)	1 (11%)	0
Protein fortification milk	4 (25%)	1 (11%)	2 (1.1%)

FEF = full enteral feeding; IUGR = intrauterine growth restriction.

**Table 2 nutrients-16-03036-t002:** Laboratory parameters of newborns with NEC, FPIAP, and idiopathic neonatal transient colitis.

	FPIAP Mean (Range Min–Max)	Idiopathic Transient Colitis Mean (Range Min–Max)	NEC Mean (Range Min–Max)
Hemoglobin (g/dL)	13.8 (8.8–22.9)	14.1 (9.7–18.3)	13.6 (8.5–19)
Platelets ×10^6^/L	353,142 (118,000–533,000)	281,428 (117,000–359,000)	240,142 (115,000–402,000)
CRP mg/L	3.8 (0–20.8)	1.6 (0–8.9)	27.2 (1–66)
White blood cells U/mL	9982 (3590–15,810)	11,192 (6450–19,770)	11,962 (2940–30,610)
Neutrophils per mm^3^	4335 (1330–6250)	2604 (942–6200)	2724 (1330–9980)
Eosinophils per mm^3^	405 (200–1000)	725 (140–2050)	277 (40–870)

## Data Availability

The original contributions presented in the study are included in the article. Further inquiries can be directed to the corresponding author.

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
