# Peer review of "Food-Protein-Induced Proctocolitis in Pre-Term Newborns with Bloody Stools in a Neonatal Intensive Care Unit"

_nutrients, 2024, doi:10.3390/nu16173036_

Round 1
Reviewer 1 Report
Comments and Suggestions for Authors
This manuscript demonstrated that CMPA should be considered in the differential diagnosis of bloody stools in premature infants, and that high serum PCT levels and low eosinophil counts might be helpful in diagnosing NEC. The findings are clinically informative. However, further information is needed to support the authors' conclusions.
Specific comments
1) The definition of transient hematochezia depicted in L190-192 should be included in the material and method section.
2) Do the figures for hemoglobin, platelets, CRP, White blood cell counts, neutrophil counts, and eosinophil counts shown in Table 1 represent the mean or median? The range, or quartile of each figure should be listed alongsides the mean or median in Table 1.
3) It is claimed that high serum PCT levels and low eosinophil counts are helpful in diagnosing NEC. However, the sensitivity, specificity, positive predictive value, and negative predictive value of serum PCT levels and eosinophil counts, or a combination of both, for diagnosing NEC should be provided, along with the results of Receiver Operating Characteristic (ROC) analysis.
4) Since eosinophilia is often observed in premature infants, reflecting hematopoiesis, it is necessary to consider the gestational ages and the number of days since birth in order to compare eosinophils. Is there a significant difference between the NCC and CMPA groups when these values are taken into account?
Minor comments
1) The full form of APLV should be provided at its first occurrence.
2) The full form of PCR should be provided at its first occurrence.
Author Response
This manuscript demonstrated that CMPA should be considered in the differential diagnosis of bloody stools in premature infants, and that high serum PCT levels and low eosinophil counts might be helpful in diagnosing NEC. The findings are clinically informative. However, further information is needed to support the authors' conclusions.Specific comments:
1) The definition of transient hematochezia depicted in L190-192 should be included in the material and method section.
R: First, we consider that the term “idiopathic neonatal transient colitis” is more appropriate. The definition of “idiopathic neonatal transient colitis” has been included in the Material and Method section (see Lines 102-104) and throughout the text, accordingly.
2) Do the figures for hemoglobin, platelets, CRP, White blood cell counts, neutrophil counts, and eosinophil counts shown in Table 1 represent the mean or median? The range, or quartile of each figure should be listed along sides the mean or median in Table 1.
R: Done. The mean and the range of each Lab parameter has been added, accordingly. In addition, to make the text clearer, two tables have been included: Table 1 with clinical characteristics and Table 2 with Laboratory parameters (expressed as mean and ranges).
3) It is claimed that high serum PCT levels and low eosinophil counts are helpful in diagnosing NEC. However, the sensitivity, specificity, positive predictive value, and negative predictive value of serum PCT levels and eosinophil counts, or a combination of both, for diagnosing NEC should be provided, along with the results of Receiver Operating Characteristic (ROC) analysis.
R: Thank you for the insightful comment. As suggested, we have conducted a ROC analysis to evaluate the predictive performance of PCT and eosinophil levels for NEC diagnosis. Specifically, we generated ROC curves and calculated AUC values. We also identified optimal thresholds and provided sensitivity, specificity, positive predictive value, and negative predictive value metrics.
According to the reviewer comment the ROC analysis with AUC value has been included into the text (lines 173-177), while we decided to not include into the text the ROC analysis for pct, also considering that PCT had not been included in the Table (Table 2 in the revised version). Accordingly, we have considered that a couple of sentences in the Discussion sentences may be confounding, since they are beyond the scope of the manuscript, which is not the description of NEC in prematures; thus, they have been deleted (Lines 166-170 of draft tracked change).
4) Since eosinophilia is often observed in premature infants, reflecting ematopoiesis, it is necessary to consider the gestational ages and the number of days since birth in order to compare eosinophils. Is there a significant difference between the NCC and CMPA groups when these values are taken into account?
R: Thanks for the comment. Really, the number of esoinophils is higher in prematures and it depends on the gestational age, being higher with lower gestational age (Juul SE et al., J Perinatol 2005;25:182-8. Han JY et al., Korean J Perinatol 2011;22:285-94; Fayon M et al.,). However, it should be noted that having performed a statistical analysis on the gestational age and number of days since birth we did not observe a difference between NEC, CMPA and ET patients, suggesting that these parameters are homogeneous among our study cohort. Nevertheless, we explored whether the gestational age or the age at symptom on set could impact the eosinophilic levels in our patient cohort, using a Pearson correlation analysis. The results showed that both gestational age (Pearson r = 0.13, p = 0.49) and the number of days since birth (Pearson r = -0.15, p = 0.43) do not have a significant inverse correlation, suggesting that the eosinophilic levels in our studies are not influenced by gestational age or children age. We have added this new information in the results section and Discussion section (see Lines 207-215).
Reviewer 2 Report
Comments and Suggestions for Authors
The manuscript is a welcome/necessary/timely addition to the scientific literature in relation to proctocolitis. However, I would invite the authors to improve both the scientific nature of the manuscript and the grammar and syntax used in the manuscript.
1. Please find below a list of references that the authors should consider adding to the reference list and also to examine the references in terms of additional useful scientific information which could be included, in different sections of the manuscript.
2. I accept that in relation to the editorial guidelines, of the journal, the number of references I have recommended is high. However, the authors do not need to include all of them if they do not wish to do so.
3. Would the authors not agree that there are too many paragraphs in the manuscript and that this makes it difficult to read in terms of "continuity"? Therefore many of the separate paragraphs should, and surely, need to be conjoined?
4. Table 1 needs to be revisited in terms of commas used for the values of the parameters described instead of "full-stops".
5. The "box plots" need to be discussed/analysed in more depth as does the statistical analysis used ; furthermore do the foregoing not require to be accounted for in a more rigorous manner?
..................................................................................................................................................
1. Qin Chen et al., Identification of diagnostic biomarks and immune cell infiltration in ulcerative colitis, Scientific Reports (2023) 13:6081
2. Weitao Hu et al., Identification of hub genes and immune infiltration in ulcerative colitis using bioinformatics, Scientific Reports (2023) 13:6039
3. Jiali Lu et al. PANoptosis and Autophagy-Related Molecular Signature and Immune Landscape in Ulcerative Colitis: Integrated Analysis and Experimental Validation, Journal of Inflammation Research 2024:17 3225–3245
4. Jakob J. Wiese et al., Myenteric Plexus Immune Cell Infiltrations and Neurotransmitter Expression in Crohn’s Disease and Ulcerative Colitis, Journal of Crohn's and Colitis, 2024, 18, 121–133
5. Jasmina El Hadad et al., The Genetics of Inflammatory Bowel Disease, Molecular Diagnosis & Therapy (2024) 28:27–35
6. Rong Huang et al., Identifying immune cell infiltration and effective diagnostic biomarkers in Crohn’s disease by bioinformatics analysis, Frontiers in Immunology 10.3389/fimmu.14:1162473. doi: 10.3389/fimmu.2023.1162473
7. Qiuyue Yuan and Zhana Duren, Continuous lifelong learning for modelling of gene regulation from single cell multiome data by leveraging atlas-scale external data, bioRxiv preprint doi: https://doi.org/10.1101/2023.08.01.551575
8. Xuhong Zhang et al., Bioinformatics Analysis of Immune Cell Infiltration and Diagnostic Biomarkers between Ankylosing Spondylitis and Inflammatory Bowel Disease, Computational and Mathematical Methods in Medicine 2023, Article ID 9065561, https://doi.org/10.1155/2023/9065561
9. Jan K. Nowak et al., Characterisation of the Circulating Transcriptomic Landscape in Inflammatory Bowel Disease Provides Evidence for Dysregulation of Multiple Transcription Factors Including NFE2, SPI1, CEBPB, and IRF2, Journal of Crohn's and Colitis, 2022, 16, 1255–1268
Comments on the Quality of English Language
As stated above the quality of the "English Language" used in the manuscript needs to be improved in terms of (i) paragraph "construction" and (ii) syntax /grammar.
Author Response
The manuscript is a welcome/necessary/timely addition to the scientific literature in relation to proctocolitis. However, I would invite the authors to improve both the scientific nature of the manuscript and the grammar and syntax used in the manuscript.
1. Please find below a list of references that the authors should consider adding to the reference list and also to examine the references in terms of additional useful scientific information which could be included, in different sections of the manuscript.
2. I accept that in relation to the editorial guidelines, of the journal, the number of references I have recommended is high. However, the authors do not need to include all of them if they do not wish to do so.
1. Qin Chen et al., Identification of diagnostic biomarks and immune cell infiltration in ulcerative colitis, Scientific Reports (2023) 13:6081
2. Weitao Hu et al., Identification of hub genes and immune infiltration in ulcerative colitis using bioinformatics, Scientific Reports (2023) 13:6039
3. Jiali Lu et al. PANoptosis and Autophagy-Related Molecular Signature and Immune Landscape in Ulcerative Colitis: Integrated Analysis and Experimental Validation, Journal of Inflammation Research 2024:17 3225–3245
4. Jakob J. Wiese et al., Myenteric Plexus Immune Cell Infiltrations and Neurotransmitter Expression in Crohn’s Disease and Ulcerative Colitis, Journal of Crohn's and Colitis, 2024, 18, 121–133
5. Jasmina El Hadad et al., The Genetics of Inflammatory Bowel Disease, Molecular Diagnosis & Therapy (2024) 28:27–35
6. Rong Huang et al., Identifying immune cell infiltration and effective diagnostic biomarkers in Crohn’s disease by bioinformatics analysis, Frontiers in Immunology 10.3389/fimmu.14:1162473. doi: 10.3389/fimmu.2023.1162473
7. Qiuyue Yuan and Zhana Duren, Continuous lifelong learning for modelling of gene regulation from single cell multiome data by leveraging atlas-scale external data, bioRxiv preprint doi: https://doi.org/10.1101/2023.08.01.551575
8. Xuhong Zhang et al., Bioinformatics Analysis of Immune Cell Infiltration and Diagnostic Biomarkers between Ankylosing Spondylitis and Inflammatory Bowel Disease, Computational and Mathematical Methods in Medicine 2023, Article ID 9065561, https://doi.org/10.1155/2023/9065561
9. Jan K. Nowak et al., Characterisation of the Circulating Transcriptomic Landscape in Inflammatory Bowel Disease Provides Evidence for Dysregulation of Multiple Transcription Factors Including NFE2, SPI1, CEBPB, and IRF2, Journal of Crohn's and Colitis, 2022, 16, 1255–1268
- Would the authors not agree that there are too many paragraphs in the manuscript and that this makes it difficult to read in terms of "continuity"? Therefore, many of the separate paragraphs should, and surely, need to be conjoined? R: Thank you for the comment, the paper has been completely revised and the paragraphs have been conjoined, as you suggested.
- Table 1 needs to be revisited in terms of commas used for the values of the parameters described instead of "full-stops". R: thank you very much for your comment, table 1 has been divided into two tables, one with the population characteristics and the other with the laboratory parameters. The full stops and commas have been adjusted.
- The "box plots" need to be discussed/analysed in more depth as does the statistical analysis used; furthermore do the foregoing not require to be accounted for in a more rigorous manner?
R: Done. The box plots have been analysed in more depth (see statistical analysis, Lines 115-124 and 162-166) and discussed (see Discussion Lines 198-215). In regard to the additional references, they refer to inflammatory bowel diseases; anyway, we considered to add 3 references (1-6-9) which we considered suitable to be added into the text.
The English language has been edited throughout the text.
Reviewer 3 Report
Comments and Suggestions for Authors
This is a retrospective study which shows as many studies have before that FPIAP is rare in extreme pre-term infants and furthermore NEC and FPIAP have very different clinical and radiological features.
I'm concerned that babies with radiological features of NEC - pneumatosis were labelled as FPIAP.
I'm not sure what the authors are referring to as 'transient haemochezia' nor do I understand when an OFC was considered.
To include all these 3 clinical phenotypes with such small numbers makes it very hard to interpret the results.
What is the clinical guidance that this study seeks to offer?
Author Response
This is a retrospective study which shows as many studies have before that FPIAP is rare in extreme pre-term infants and furthermore NEC and FPIAP have very different clinical and radiological features. I'm concerned that babies with radiological features of NEC – pneumatosis were labelled as FPIAP. I'm not sure what the authors are referring to as 'transient haemochezia' nor do I understand when an OFC was considered. To include all these 3 clinical phenotypes with such small numbers makes it very hard to interpret the results. What is the clinical guidance that this study seeks to offer?
R: Thank you for the comment which gives us the opportunity to better explain our results and their interpretation. In regard to the reviewer’s concern, in clinical practice the finding of intestinal pneumatosis may be observed also in clinical entities different by NEC, as it has been described for FPIES (Hernández-Almeida P et al., Food protein-induced enterocolitis syndrome with pneumatosis intestinalis in an exclusively breastfed infant: A case report and literature review. Clin Case Rep. 2022 Nov 23;10(11):e6520.). However, intestinal pneumatosis has also been described in FPIAIP and initial misdiagnosis of NEC may occur in clinical practice and it has been just described (Liu H, Turner TWS. Allergic Colitis With Pneumatosis Intestinalis in an Infant. Pediatr Emerg Care. 2018 Jan;34(1):e14-e15.). Accordingly, we have added a bit in the Discussion to better clarify this issue (Lines 219-229)
We thank the reviewer for the comment regarding the definition of transient ematochezia, which has been more properly defined as “idiopathic neonatal transient colitis”, which has been described for term infants. The definition of “idiopathic neonatal transient colitis” has been included in the Material and Method section (see Lines 102-104) and throughout the text, accordingly.
We also explicit the time for the oral food challenge (see Line 98)
We know that the number of patients were small, and this is a study limitation; however, the findings of the present study offer to the clinicians, especially to neonatologists and allergologists who take care of prematures, the novel message regards the important role of the oral food challenge to differentiate between transient idiopathic neonatal colitis and FIAIP also in preterm newborns. It is pivotal to make a correct diagnosis, avoiding unnecessary and prolonged exclusion diet in case of transient idiopathic colitis.
Round 2
Reviewer 1 Report
Comments and Suggestions for Authors
The authors appropriately revised the manuscript
Reviewer 3 Report
Comments and Suggestions for Authors
The authors are to be congratulated on making the changes that have been requested, this makes the article much clearer to the reader.
There are important and well-made clinical points within the article albeit not novel nor large in numbers.
Comments on the Quality of English LanguageFew spelling issues - 'rouled' not ruled as an example